# Thermal and Photo Sensing Capabilities of Mono- and Few-Layer Thick Transition Metal Dichalcogenides

**DOI:** 10.3390/mi11070693

**Published:** 2020-07-17

**Authors:** Andrew Voshell, Mauricio Terrones, Mukti Rana

**Affiliations:** 1Division of Physics, Engineering, Mathematics and Computer Sciences and Optical Science Center for Applied Research, Delaware State University, Dover, DE 19901, USA; abvoshell13@students.desu.edu; 2Department of Physics, Chemistry and Materials Science & Engineering, Pennsylvania State University, University Park, PA 16802, USA; mut11@psu.edu

**Keywords:** transition metal dichalcogenides, semiconductor, monolayer, temperature sensor, photodetector, band gap, thermocouple, broadband

## Abstract

Two-dimensional (2D) materials have shown promise in various optical and electrical applications. Among these materials, semiconducting transition metal dichalcogenides (TMDs) have been heavily studied recently for their photodetection and thermoelectric properties. The recent progress in fabrication, defect engineering, doping, and heterostructure design has shown vast improvements in response time and sensitivity, which can be applied to both contact-based (thermocouple), and non-contact (photodetector) thermal sensing applications. These improvements have allowed the possibility of cost-effective and tunable thermal sensors for novel applications, such as broadband photodetectors, ultrafast detectors, and high thermoelectric figures of merit. In this review, we summarize the properties arisen in works that focus on the respective qualities of TMD-based photodetectors and thermocouples, with a focus on their optical, electrical, and thermoelectric capabilities for using them in sensing and detection.

## 1. Introduction

Two-dimensional (2D) materials have been heavily studied over the past two decades since the first synthesis of graphene in 2004 [1]. By separating the weak Van der Waals structure of different bulk materials, researchers have reduced layered materials to their limit of a single atomic layer. This development has been shown to cause new characteristics of materials in their single- and few-layer form, creating an entirely new class of material with new structural, electrical, and optical properties [2,3,4]. The layered nature of these materials also allows layer-dependent properties to be exploited, showing altered optical and electrical properties which can be tailored for specific applications as well as developing heterostructures with multiple materials for enhanced performance.

Of these layered materials, semiconducting transition metal dichalcogenides (TMDs) are of the form MX_2_, which consists of a transition metal M (Mo, W, etc.) sandwiched between two chalcogenide atoms X (S, Se, Te, etc.), with a thickness of less than 1 nm [5]. The atomic diagram of TMDs is shown in Figure 1. These materials have been gaining much attention recently in various research field due to the fact of their unique properties [6]. Bulk TMDs are formed by stacking the layers. Weak Van der Waals forces hold the TMDs layers together, and they can be easily separated to form monolayer or few-layers [7]. Monolayer TMDs can have much different properties than their bulk counterparts. For example, monolayer TMD materials have a direct band gap, opposed to few-layer and bulk TMDs, giving monolayer TMDs very strong light absorption and photoluminescence properties [8,9]. The band gap characteristics also allow them to be used in many semiconducting as well as optoelectronic applications such as photodetection, thermoelectric generation, and transistors [10]. Furthermore, TMDs can alter their properties by changing the strain [11], defect engineering [12], and alloying of materials [13]. The semiconducting properties and tunability allow them to be useful in both photodetection and thermoelectric generation.

Thermal sensing is very important in research, industry, military, and consumer applications and have been developed for hundreds of years. The ability to accurately measure or sense temperature change is dependent on the application and material which can be used for that particular case. Recent advances in material science and electronics have given highly sensitive contact and non-contact methods of measuring a very wide range of temperatures [14,15]. However, many of these materials are bulky, toxic or expensive. For example, short wave infrared detection is commonly performed by mercury cadmium telluride (MCT) and InGaAs thermal photodetectors, but MCT is very toxic and expensive to produce and cool [16], and InGaAs lacks wide broadband photodetection [17]. However, some TMD optoelectronics offer the capability of measuring some of the spectral range of these materials with drastically lower cost and toxicity while also not requiring cooling to the system [18]. Also, since some of the 2D TMDs have good thermoelectric properties, namely, its low thermal conductivity compared to graphene [19] and high power factor [20], they have the possibility of being utilized in contact-based thermal sensors as well [21]. 

The ability to minimize the size of sensing devices has become useful for both contact and non-contact-based sensing [16]. Ultrathin and flexible devices can be used in novel research and industrial applications in the technology world as devices become smaller and more advanced. Due to the tunability, compactness, low-cost, flexibility, and non-toxic nature of many of these materials, 2D TMDs can have a great impact on the development of thermal and photo sensing devices and mechanisms. In this review, we look at the optical, thermal, and electrical capabilities in TMDs and heterostructures and show their usefulness in novel optoelectronic and thermocouple based systems.

## 2. Theory

### 2.1. Non-Contact

There are various methods of non-contact thermal sensing, but many of them utilize the blackbody radiation of the object in question. The peak light radiation is proportional to the temperature of the object described in Planck’s law of blackbody radiation, given by the equation [23]:(1)$$B_{\nu }=\frac{2h\nu^{3}}{c^{2}}\frac{1}{e^{\frac{h\nu }{kT}}-1}$$
where *B_ν_* is the spectral radiance at a given temperature, *h* is Planck’s constant, *c* is the speed of light, *k* is the Boltzmann constant, *v* is the frequency of the electromagnetic radiation, and *T* is the temperature of the body. Figure 2 shows how the peak spectral radiance changes with the temperature of a blackbody.

The photodetection capabilities of a semiconductor is mostly influenced by the size of the band gap of the material [24]. If incident photons on a semiconductor material are of an energy greater than or equal to the bandgap of the semiconductor, then an electron–hole pair is generated, and electrons from the valence band move to free states in the conduction band, where they can move as free electrons, generating a photocurrent [25]. Many properties and development methods can affect the band gap of the semiconductor, such as doping, strain, crystallinity, and defects [26,27].

As Equation (1) and Figure 2 show, peak spectral radiance blue shifts to shorter wavelengths as the temperature increases. The most emissive wavelength of light is simply shown by Wien’s displacement law, given as [28]:(2)$$\lambda_{peak}=\frac{b}{T}$$
where, *λ_peak_* is the peak wavelength of light emitted by a blackbody of temperature *T*, and *b* is Wein’s displacement constant whose value is given as 2.8977 × 10^−3^ m-K. Therefore, in order to provide the highest sensitivity of a photodetector for blackbody measurement, one would choose a sensor tailored to measure the peak emission of the blackbody in question. Table 1 displays some emitted photon energies by the temperature of various common applications. Thermal sensing via photodetection is most commonly used and described in the wavelength range of 8–12 μm, since this accounts for the temperature distinguishing between the blackbody radiation of humans (~310 K) and background (~290 K) with good atmospheric transmission.

Transition metal dichalcogenides (TMDs) offer the capability of being tuned to a variety of band gaps due to the layered nature of the semiconductor. This capability allows a TMD-based photodetector to be tailored to have maximum absorption for peak blackbody emission, allowing high-performance devices to be made for specific cases. The low cost, ability to alloy, and simplicity of growth allows a wide variety of applications for optoelectronics, light detection, and optical assembly [29].

### 2.2. Contact

Contact-based method of thermal sensing requires the sensor to be in direct contact with the object in question. Many different mechanisms involve contact thermal sensing, but the most common is the thermocouple, and is used commonly in industrial, automotive, and consumer applications. Thermocouples utilize the thermoelectric effect, where the temperature change of a junction of two different materials, such as different metals or semiconductors, create an electrical response given as [30]:(3)V=SΔT
where *V* is voltage, Δ*T* is the temperature difference across the junction, and *S* is the Seebeck coefficient, which describes the induced thermoelectric voltage over a temperature difference of the material. When a temperature gradient is generated in thermoelectric materials (usually semiconductors), electrons and holes with high thermal energy at the hot end diffuse to the cold end, creating a potential difference to power an external load. Figure 3 shows the basic mechanism of a thermoelectric generator.

Although metal thermocouples are commonly used due to the fact of their cheap cost, highly sensitive thermocouples are made with an n- and p-doped semiconductors. In order to determine the ability of a thermocouple to have good temperature-to-electric conversion efficiency, it is ideal to maximize the figure of merit ZT, given by [31]:(4)$$ZT=\frac{S^{2}\sigma }{\kappa_{e}+\kappa_{ph}}T$$

In which, *σ* is the electric conductivity, *κ_e_* is the electron thermal conductivity, and *κ_ph_* is the lattice thermal conductivity. In order to optimize the figure of merit, it is ideal for a semiconductor to have a high Seebeck coefficient and electrical conductivity, while minimizing the thermal conductivity of the material. Conventional thermoelectric materials have a ZT value of about 0.7–1, while very good devices have a ZT value of about 1.5–2.5, with very few materials achieving above 2.5 [32]. Very few outliers achieve a figure of merit above 3, although some reports have shown a ZT value as high as 6 in some cases [33].

## 3. TMD-Based Photodetectors

Broadband photodetection is a method of thermal sensing with TMDs as well as many different semiconductor materials for broadband temperatures. TMDs are a class of semiconductors that have energy band gap range of about 1.2 eV to 2.1 eV, which corresponds to a wavelength of 590–1033 nm or a temperature peak of 2800 K–4900 K. The visible and NIR range of common TMD-based photodetectors make these detectors possible to sense high temperature objects such as—thrusters, high temperature metals, and some devices used in solar applications. Also, the ability to tune and extend the band gap by altering the number of layers can make these materials attractive for novel thermal sensing applications.

### 3.1. Mono- and Few-Layer Photodetectors

When a TMD is a monolayer, the material is a direct band gap semiconductor, and it becomes an indirect band gap semiconductor when in few layers [34]. At the monolayer level, the band gap of TMDs range from about 1.2–2.1 eV, which limits photodetection in the UV, visible, and NIR range, with varying performance based on the quality and type of material [35,36]. Although various reports have shown high performance devices of these materials intrinsically, the use in thermal photodetection is limited to very high temperature detection, as the detection is mainly in the visible range, with some detection in the NIR range.

The most common TMD for photodetection is MoS_2_, with many studies focused on its photodetection capabilities. Lopez-Sanchez et al. [37] developed a highly sensitive MoS_2_ photodetector with a spectral range of 400–680 nm with a photoresponsivity of 880 AW^−1^ with a drain-source voltage of 8 V and a backgate voltage of −70 V, shown in Figure 6a. Although this could only be used for very high thermal sensing in a small wavelength range among other novel applications, this photodetector far surpassed previous monolayer MoS_2_ photodetectors, and opened the window to new development in TMD photodetector technology. However, although the photodetector in their work was highly sensitive, the presence of trap states greatly reduced the response time of the device, with a field-effect mobility µ of about 4 cm^2^V^−1^s^−1^. Yu et al. [38] resolved this issue by using (3-mercaptopropyl)trimethoxysilane (MPS) in thiol chemistry to repair sulfur vacancies in MoS_2_ monolayers in both top-side and double-side methods. The group reported a double-side-treated MoS_2_ monolayer mobility of ~81 cm^2^V^−1^s^−1^, higher than any reported experimental mobility at the time, with theoretical maximums of over >400 cm^2^V^−1^s^−1^. The benefit of this method is the capability to be used in other TMD materials as well. For example, WS_2_ has a phonon-limited electron mobility of up to 1100 cm^2^V^−1^s^−1^, much higher than MoS_2_ [39]. Cui et al. developed a monolayer WS_2_ field-effect transistor and enhanced electron mobility using the abovementioned thiol chemistry alongside adding an ultrathin dielectric layer between WS_2_ and SiO_2_, further reducing the density of charge traps. Both methods allowed them to achieve a mobility of 83 cm^2^V^−1^s^−1^ at room temperature, and 337 cm^2^V^−1^s^−1^ at low temperatures. Other strategies can also further enhance the electron transport in monolayer TMDs, such as strain engineering and doping, for further improvement in the electron mobility of TMDs, leading to faster, high-performance photodetectors [40,41].

Optical absorption of mono and few-layer TMDs have three prominent peaks. Two are due to the direct transitions at the K point of the Brillouin zone from generation of A and B excitons [42]. The other one is a broad peak due to singularities in joint density of states between the first valence and conduction band near the Γ point of the valence band [43]. Trion’s also contribute a small amount to optical absorption, with a small absorption tail with a slightly lower energy than the A exciton peak [44].

Several methods have been imposed to enhance the absorbance of TMD’s for better photodetector performance. Butun et al. [45] incorporated Ag plasmonic nano-discs on a monolayer WS_2_ sample by e-beam lithography. By using 100 nm diameter disks, they were able to increase the absorption at the absorption peak (610 nm) up to over 20%, a 2.5-fold increase in optical absorption compared to the bare WS _2_ sample. Huo et al. [46] used a similar approach, but used hexagonal titanium nitride nano-disc array, which showed near-perfect absorption (>98%). Although there was a slight decrease in absorption with the MoS_2_ layer at the peak as opposed to only the nano-discs, the addition of MoS_2_ broadened the absorption band, allowing better broadband absorbance from 475 nm to 772 nm. In order to extend light absorption to longer wavelengths, studies have also used more layers of MoS_2_ to lower the band gap of the material [47]. Park et al. [48] used chemical exfoliation to develop MoS _2_ photodetector with thicknesses of 5, 10, and 25 nm, equating to approximately 8, 17, 42 layers, respectively. They found a linear increase in absorption with increase in film thickness, with the 25 nm photodetector outperforming others from 400–1600 nm wavelength range (See Figure 4). The increased thickness led to a larger number of defects, which not only increased photocurrent and dark current, but also increased the response time of the device. The increase in these values are due to the increase of defects present in thicker films, which causes a further reduction in band gap [49]. Furthermore, the group implemented a layer of plasmonic silver nanoparticles (AgNPs) in order to concentrate incident near infrared (NIR) light, resulting about three times stronger absorption than the plain film. They were able to achieve the responsivity of 0.881 mA/W and a detectivity of 1.28 × 10^9^ Jones.

Molybdenum ditelluride (MoTe_2_) has the lowest bandgap energy among the un-altered TMDs (by layer), with a direct gap of about 1.2 eV. The material’s high absorption and good electrical capabilities made it a good contender for NIR and short-wave infrared (SWIR) photodetectors, which is useful for telecommunications as well as thermal sensing at high temperature. Huang et al. [50] displayed a sensitive VIS-NIR photodetector by mechanically exfoliating 2H-MoTe_2_ onto a SiO_2_/Si substrate. The detector was enhanced by the photogating effect. When electrons or holes are trapped in localized states in the valence and conduction band of the material, this leads a voltage to be applied to the device. This way, it extends the lifetime of the electrons, thus increasing the photoconductive gain of the device and this effect is known as photogating effect [51]. Through their work, they were able to achieve a responsivity of about 24 mAW^−1^ and a detectivity of 1.3 × 10^9^ cmHz^1/2^W^−1^, with a backgate voltage of 10 V, with a detected photo response of 600–1750 nm. The group noted that a higher backgate voltage allowed for a higher responsivity, but increased the dark current, thus lowering the detectivity of the device.

In the mid and long wavelength range, there have been some demonstrations of few-layers of TMD photodetectors for sensing these ranges of heat. Thermal sensing in the mid-long wavelength infrared region is regarded as “thermal radiation” and refer to radiated light in the range of approximately 3–15 µm, or a temperature of 193–966 K. Mid-wavelength infrared thermal sensing around 3–5 µm is useful for vehicle engine and missile heat sensing, but the 5–7 µm range is of little use to photodetection due to high atmospheric absorption [52], while long-wave infrared (LWIR) can be used for distinguishing human temperatures with background, allowing night vision sensing. Further development of inducing defects into multi-layered TMDs have shown to be promising for further lowering the band gap of the material for mid-long IR absorption.

Very few studies have shown that a few layer-thick TMDs are capable of extending their optical absorption to the mid and LWIR range. However, studies have shown that altering the ratio of Mo and S atoms in few layered MoS_2_ samples allowed great reduction of the band gap, allowing longer wavelength light absorption. Xie et al. [53] used pulsed laser deposition of MoS_2_ to alter the ratio of sulfur and molybdenum atoms in grown layers. This way they altered the band gap of the film. The prototype photodetector showed a broadband detection response ranging from 445 nm to 9.5 µm. This wide range allows for mid-infrared and some LWIR thermal sensing from an uncooled photodetector, with a responsivity of 21.8 mAW^−1^ at 7.79 µm (see Figure 5), which is higher than any other realized room-temperature photodetectors currently used at this wavelength. The realized detector was sensitive from visible light to 9.5 µm, making it the widest band room-temperature photodetector to date.

### 3.2. Heterostructure Photodetector Devices

Although properties of various TMDs can be altered by material engineering, the development of heterostructures and interfaces have given new capabilities to photodetection for possible thermal sensing applications. Since many 2D materials, including TMDs, are held by weak Van der Waals forces, complex heterostructures can be made by manually stacking layered materials, resulting in many possibilities in optical devices.

Different layered materials can offer different capabilities in the heterostructure photodetector. For example, graphene has been used commonly with TMDs in order to create layered electrodes on the device which are easy to use and have fast electrical transport, while maintaining the atomic-level thinness of the device, as shown in Figure 6b [54]. Furthermore, the stacked graphene–TMD heterostructures can even cause synergistic benefits to photodetectors such as modulation of photoresponsivity and enhanced light–matter interactions [55]. Table 2 shows a summary of various TMD-based photodetectors and their figures of merit.

Long et al. [57] demonstrated a TMD and graphene-based heterostructure using MoS_2_ and WSe_2_ for a p-g-n diode for broadband photodetection capabilities. Figure 6c shows the top and side view of the MoS_2_/WSe_2_ heterostructure. The device utilizes the broadband photodetection of graphene to extend the photodetection past the band gap limitations of MoS_2_ and WSe_2_, for a VIS-NIR photodetection range of around 400–2400 nm. While the TMDs and graphene both generate photocurrent in the visible range, longer wavelength light is not absorbed by the TMDs resulting in only graphene providing photocurrent for the infrared range, resulting in a lower responsivity of the device in that range. However, the group reported a responsivity 106AW^−1^ and detectivity of up to 1015 Jones in the visible range, while in the infrared region they found a responsivity and detectivity of about 1AW^−1^ and 10^11^ Jones, respectively.

The ability to scale up simple p-n junction design and compatibility with complementary metal-oxide-semiconductor (CMOS) technology is highly sought for moving TMD-based photodetectors into applications outside of research. Dhyani et al. [59] demonstrated a Si/MoS_2_ heterojunction using wafer-scale processes which can easily be scaled and reproduced for photodetector arrays. The photodetector exhibited a response of ~8.75 A/W (at 580 nm) and detectivity of 1.4 × 10^12^ Jones, with a fast response time of 10 µs. The formation of the heterojunction also has an inherently higher depletion region which becomes stronger under reverse bias. The process gives an increased responsivity for higher wavelengths (>500 nm) when compared to only MoS_2_, leading to its use in broadband photodetector applications. Other groups have also used TMD-silicon heterostructures for high-speed, high-sensitivity photodetectors. Wu et al. [60] also developed a TMD/Si heterostructure, but using few-layer WS_2_/Si with a type-II band alignment formed in situ. Furthermore, they formed a 4 × 4 array with similarly high reproducibility and stability. The type-II band alignment allows broadband absorption of up to 3043 nm with a responsivity of 224 mA/W and detectivity of 1.5 × 10^12^ Jones.

Xue et al. [63] demonstrated a few-layer-thick MoS_2_/WSe_2_ heterojunction array by performing a thermal reduction sulfurization process. Using their two-step chemical vapor deposition approach, the group was able to create arrays of MoS_2_/WSe_2_ stacks with well-defined interfaces, allowing the possibility of building various Van der Waals heterojunctions in a large scale. The MoS_2_/WSe_2_ devices in the report showed a responsivity of 2.3 AW^−1^ over the visible spectrum, with highly stable and fast switching behavior. The array is also flexible, which gives some niche applications, allowing some novel photodetectors for possible high-temperature blackbody sensing. Lee et al. [64] also showed the enhanced photodetection with p-n heterojunction, shown by using MoS_2_/WSe_2_ sandwiched between graphene electrodes. The combination of these qualities and techniques allow the possibility for sensitive, large scale, and easily producible photodetector arrays. A diagram of their heterostructure device is shown in Figure 6d.

Flexible devices have also sparked interest in various areas such as wearable electronics and non-static or curved systems. The mechanical properties and band-selective capabilities of TMD’s allow them to be a great candidate for flexible optoelectronics [65,66]. Several groups have used nanosheets with graphite on paper with TMD nanosheets to make a simple, flexible photo-detecting heterostructure with good performance [67,68]. For example, Pataniya et al. [61] displayed a simple photodetector using ultrasonically exfoliated WSe_2_ nanosheets on pencil-drawn graphite. The sensor was made on a paper substrate, with a responsivity of 6.66 mA/W and detectivity of 1.94 × 10^8^ Jones, with a response time of 0.8 s and broadband sensitivity. The simple design and low-cost materials, along with the flexible structure, make the device useful for many possibilities, such as visible and infrared photodetection of blackbody radiation.

Other flexible devices have utilized the strain effects on monolayer TMD’s in order to develop a flexible heterostructure with increased performance. Zhang et al. [62] developed a flexible photodetector based on a p-CuO/n-MoS_2_ heterojunction, and utilized the piezo-phototronic effect to enhance performance with increased strain. The coupling among semiconductor, piezoelectricity, and photoexcitation can cause a broadening of the depletion region at the heterostructure interface, causing the piezo-phototronic effect [69]. The group showed that with 0.65% tensile strain, a photocurrent enhancement of up to 27 times than that of strain-free conditions, and a detectivity of up to 3.27 × 10^8^ Jones can be achieved.

## 4. TMD-Based Thermocouple/Thermoelectric Devices

High-quality thermocouple devices are needed for a variety of applications. These devices utilize thermoelectric effect that are used for thermal sensing in contact mode. Since metal thermocouples tend to have poor performance, semiconductor thermocouple devices can be used in specific situations requiring ultra-compact or thin areas. In order to maximize the capabilities of TMDs as a thermocouple, the Seebeck coefficient and electronic conductivity are maximized, while thermal conductivity is minimized, in order to produce the highest possible figure of merit. The electrical conductivity, Seebeck coefficient, and thermal conductivity are not independent, which makes it difficult to optimize one parameter without affecting the other parameters. Hence, substantive efforts on improving ZT focus on the development of new TE materials and/or their structural optimizations [70]. The intrinsic quantum size effects make the physical factors defining ZT free from the interdependence mentioned above [21,71].

Monolayer and nano-scale materials have been shown to have promising capabilities in thermocouple devices. For example, graphene has been shown to have a high Seebeck coefficient and electronic conductivity, but is limited by its extremely high thermal conductivity of about 5300 Wm^−1^K^−1^ [72,73]. TMD monolayers on the other hand, have shown thermal conductivity in few layer materials of about 52 Wm^−1^K^−1^ for exfoliated monolayer MoS_2_, a vast improvement from graphene in terms of 2D thermoelectric materials [74]. Furthermore, theoretical calculations and other materials have shown the possibility of much lower thermal conductance values [75]. This along with the other groups working to maximize the power factor makes TMD’s a great contender for thermoelectric generation.

It is also important to note the many factors that can alter the characteristics of TMD-based thermoelectric devices. Strain effects [76], doping [77], layer number [78], fabrication methods, and measurement techniques can all be factors in theoretical and experimental properties [79]. This allows the capability of using and combining engineering and measurement methods to develop devices with higher performances or tuning a device for specific needs in thermal sensing. The resulting devices can possibly produce higher thermoelectric figure of merit values at certain temperatures than common bulk thermoelectric materials, shown in Figure 7.

In order to create an efficient TMD-based thermocouple, the thermoelectric figure of merit and temperature range are the deciding factors in such devices. Huang et al. [81] did theoretical thermoelectric performance calculations using a 2D ballistic transport approach for multiple monolayer TMD’s from 60–500 K. The study found a temperature-dependent figure of merit for MoS_2_, MoSe_2_, WS_2_, and WSe_2_ in both n and p-doped forms. The calculated figure of merit ranged from 0.2–0.55 at room temperature (300 K), and about 0.5–1.1 at 500 K. The highest figure of merit at room temperature was of p-MoS_2_ at 0.55, while WSe_2_ showed the highest value at 500 K of 1.1. While MoS_2_ showed a high Seebeck coefficient at high temperatures, the material has a high thermal conductivity from its large Debye temperature, which is due to its low atomic mass [82].

Wickramaratne et al. [71] theoretically investigated the electronic properties and the thermoelectric performance of bulk and 1–4 monolayers of 4 different TMD materials doped with p and n type impurities: MoS_2_, MoSe_2_, WS_2_, and WSe_2_. By using density functional theory with spin-orbit coupling, they reported n-type ZT values ranging from 1.91–2.39 at 300 K for the compounds, with 2L MoSe_2_ being the highest figure of merit, and an improvement of a factor of 7.5 over the bulk values. For p-type ZT, the maximum value was of 1.15 for 2L-MoS_2_, an improvement of a factor of over 14 from bulk, with a range of 0.62–1.15 for all materials. The group also calculated the Seebeck coefficient of the compounds at the reduced Fermi energy correlating to peak figure of merit and power factor for the materials. The result was a Seebeck coefficient at maximum ZT of 287 μVK^−1^ for n-type MoSe_2_, and 245.6 μVK^−1^ for MoS_2_, while at maximum power factor was a best S value of 171 μVK^−1^ for n-WSe_2_ and 90.4 for p-MoS_2_. The application of these materials may affect the optimization of realized materials to achieve the desired figure of merit or power factor for a specific use.

Ge et al. [83] investigated the electronic structures and transport properties of the 1T″ phase MX_2_ (M = Mo, W; X = S, Se, Te) using first-principles calculations with Boltzmann transport theory. They found a direct band gap at the K point for all molybdenum TMDs, and among these three cases, the hole carrier mobility of MoSe_2_ was far higher than the other compounds, estimated to be as high as 690 cm^2^/V-s at room temperature. For this reason, along with the weak electron–phonon coupling of 1T” MoSe_2_ showed outstanding transport performance of the compound. They also evaluated the Seebeck coefficient of MoSe_2_, which was as high as ∼300 μV/K at room temperature. In their work, the highest thermoelectric power factor of MoSe_2_ was found to be 10.2 × 10^3^ µW/mK^2^ at 200 K, with a power factor of about 6 × 10^3^ µW/mK^2^ throughout the range of 100–500 K. This makes the 1T” phase MoSe_2_ promising thermoelectric material with capabilities in thermal sensing for ultrathin applications, as it is comparable to the more common 1H, but has lower thermal conductance of about 10.7 Wm^−1^K^−1^, which can result in higher ZT values [84].

Single crystalline 2D MoS_2_ layers have exhibited a significantly large value of Seebeck coefficient of ~10^5^ µVK^−1^ via tuning of an external electric field [85]. Figure 8a shows an image of a tested MoS_2_ thermoelectric generator from their work. Mechanically-exfoliated few layer 2D WSe_2_ layers also presented similar increase in performance, measured via ionic gating by Yoshida et al. [86]. The group optimized electric field tuning and found an increase of Seebeck coefficient of one order or magnitude. 

Pu et al. [87] also developed and studied thermoelectric generation of monolayer MoS_2_ and WSe_2_ devices. In many works, the size of 2D TMD flakes used have an area of about ≤10 µm^2^, which made them unsuitable for determination of the thermoelectric properties. To solve this problem, the group incorporated large-area CVD-grown 2D MoS_2_ and 2D WSe_2_ monolayers and their thermoelectric properties were determined in a FET configuration. A FET channel length of 400 µm was maintained to ensure reliable thermoelectric measurements via uniformly large temperature gradient. From this, large Seebeck coefficient (|S| >200 μVK^−1^) and power factor (>200 μWm^−1^ K^−2^) were observed in 2D MoS_2_ and 2D WSe_2_. 

Defect engineering has also been shown to alter and enhance thermoelectric performance in TMD’s. Thermal conductance has been shown to reduce under monolayer TMD samples with higher number of defects than monocrystalline samples. Yarali et al. [88] compared monolayer crystalline MoS_2_ with CVD-grown MoS_2_ and found that CVD-grown samples showed a decreased thermal conductivity by up to 50% due to their low angle grain boundaries. For atomic-level defects, Chen et al. [89] showed through theoretical simulations of monolayer MoS_2_ that decreased phonon group velocity by defects, as well as phonon localization around lattice defects, resulted in a decrease in thermal conductivity of up to 75%, from ~42.2 Wm^−1^K^−2^ to 10.5 Wm^−1^K^−2^. However, thermoelectric power factor is negatively affected due to the larger scattering in the conduction band of defect-induced monolayer MoS_2_, as Adessi et al. showed [90]. The group found through theoretical calculations that through p-type doping, electronic transport is not very affected by sulfer vacancies, resulting in a ZT nearly independent of both vacancies and length of the system. However, for n-type doping, a drastic reduction of ZT was shown with change of length, thus very low vacancy concentration is needed to maintain significant ZT values.

It has been reported that utilizing metal chalcogenides and 2D TMD hybrids with other functional materials such as graphene or reduced graphene oxide (rGO) can achieve better TE properties compared to mono-component nanostructures. Doing so, it had been reported to achieve decoupling between phonon scattering and electron transport by using hybrid interfaces, allowing reduced thermal conductivity without altering the electrical conductivity, resulting in a higher figure of merit [91]. Wang et al. [92] incorporated metallic 1T phased 2D MS_2_ (M: Mo, W) layers into rGO. They obtained a maximum power factor of 15.1 and 17.4 µWm^−1^K^−2^ in rGO/MoS_2_ and rGO/WS_2_, respectively. Oh et al. [93] displayed a TMD/graphene hybrid device, developing a MoS_2_/Graphene nanoribbon heterostructure with enhanced electrical conductivity and power factor of 700 S/m and 222 µWm^−1^K^−1^, respectively. Although graphene thermal conductance is very high, it can be significantly reduced from about 3200 Wm^−1^K^−2^ to ~80 Wm^−1^K^−2^ by using nanopatterned graphene such as graphene nanoribbons (GNR), which would result in very high ZT values [94].

2D WS_2_/poly(3,4-ethylenedioxythiophene) (PEDOT): poly(styrenesulfonate) (PSS) hybrids were also developed by sonicating 1T phase 2D WS_2_ flakes into an aqueous solution of PEDOT: PSS [95]. They reported that the presence of PEDOT: PSS chains in WS_2_ reduced the energy barrier within adjacent WS_2_ flakes and also facilitated the transport of charge carriers. A TE power factor of 45.2 µWm^−1^k^−1^ was achieved in these hybrids, which was four times higher than their pure WS_2_. Li et al. [96] performed a similar study using PEDOT:PSS with MoSe_2_ nanosheets, resulting in an power factor of 48.6 µWm^−1^k^−1^, and improvement compared to individual polymer or MoSe_2_ devices. The combination of such studies can be used with others shows that the 2D inorganic/polymer composites can be easily fabricated, and can have a large effect on the capabilities of thermal sensors or other thermoelectric applications.

TMD/TMD heterostructures have also utilized the above-mentioned benefits of interlayer interactions for increased thermoelectric performance. Wu et al. [97] investigated thermoelectric properties of bulk and bilayer 2D MoS_2_/MoSe_2_ heterostructures using density functional theory in conjunction with semiclassical Boltzmann transport theory. They predicted that the bulk 2D heterostructures could considerably enhance the thermoelectric properties as compared with the bilayer MoSe_2_. The enhancement originates from the reduction in the band gap and the presence of interlayer van der Waals interactions. They reported the variation of Seebeck effect and thermal conductivity at various temperature between 300 K to 1200 K with doping concentration and found that p-type heavily doped one had shown enhanced TE properties as compared to its n-type counterpart. At 300 K, they found the value of relaxation time dependent power factor ($S^{2}\sigma /\tau \;$) to be 1.23 × 10^11^ Wm^−1^K^−2^s^−1^). At the same temperature they reported the value of Seebeck coefficient to be ~400 μV/K for both in-plane and cross-plane conditions at a doping concentration of 10^19^ cm^−3^. Thus, for TMD/TMD heterostructures to be effectively utilized, several layers should be incorporated to the structure. 

TMD nanoribbons have also pushed thermoelectric generators to further capabilities by layering different interfaces of devices. Layer mixing has shown to both improve power factor of 2D TMD-based thermoelectric devices and reduce the thermal conductance of the device [99]. Ouyang et al. [100] showed via first principles calculations that hybrid interfaces drastically reduced thermal efficiency in armchair nanoribbons, resulting in extremely high ZT. Through their calculations, they found a MoS_2_/MoSe_2_ armchair structure with an optimized ZT value of 7.4, higher than any other thermoelectric material previously calculated or fabricated. Figure 8b shows a diagram of a MoS_2_ nanoribbon armchair thermoelectric generator design [98].

Table 3 shows a summary of the thermoelectric properties of various TMD-based devices and theoretical calculations. Although there is no definitive “best” TMD device, the tunability of many figures of merit in such devices allow the ability to tailor a device for specific applications and have comparable or better thermoelectric properties than currently used bulk materials.

## 5. Challenges and Future Efforts

Although many high-performance TMD-based sensors have been made, many challenges face the future of their use in commercial or industrial use, and the ability to be competitive with current standard devices. First, the synthesis and growth of large, high-quality samples is difficult. Although mechanical and chemical exfoliation techniques cheap and effective for research purposes, they rarely form large area samples and cannot properly control the layer number. CVD is a common approach for growing TMD’s, with the capability of single mono or few-layer crystals up to centimeter size [102]. However, this requires a long development time for production, and can allow debris and non-uniform size of samples. Another challenge is the current transfer methods of TMD’s. The transfer of samples usually involves etching the oxide layer underneath and physically transferring to other substrates or semiconductors, which can cause dirt or particulates to become trapped underneath samples, hindering their adhesion and performance. The transfer can also cause changes in strain for CVD-grown samples, as intrinsic tensile strain in caused during the growth process, which can cause dissimilarities before and after transfer [103]. Until the growth can become automated or utilize CMOS methods such as photolithography, the development of large arrays is reduced to single diode or very small arrays. Also, the degradation from various elements such as presence of oxygen in water and UV irradiation for the long-term use of TMD sensors [104]. The incorporation of new growth or preservation methods may be required for durable photodetectors and thermocouples [105].

As for TMD photodetectors and heterostructures, there is much room for improvement in terms of response time sensitivity. Since absorption of monolayer TMD’s are rather low (<10%), methods must be used to improve the light absorption such as material quality, heterostructure development, and doping. For the sensitivity of photodetector devices, the rather high band gap of mono- and few-layer materials limits TMD’s to mostly visible and near-infrared detection, and increasing the number of layers for longer wavelength absorption can drastically reduce the sensitivity, mobility, and other properties. The inclusion of nanoparticles and quantum dots, however, can greatly improve broadband absorption while preserving and even improving the sensitivity and response time of the device [106]. For low carrier mobility, several options, such as high-quality growth and the abovementioned thiol chemistry and high-ƙ dielectrics can greatly improve performance. Ultimately, for competitive photodetectors, a combination of the abovementioned methods will be required for maximum performance.

Transition metal dichalcogenides (TMD)- based thermocouples succumb to the same limitations in growth and quality of materials as photodetectors, so the progression in the development in fabrication strategies is important to high thermoelectric performance. Also, the contact between source and drain electrodes with TMD thermoelectric materials inhibits the performance of thermoelectric devices [107,108]. Several strategies can be used to address this. For example, low work function metals, such as scandium and aluminum, can provide ohmic contacts to MoS_2_, allowing thermoelectric devices with enhanced performances. The combination of metal selection with proper doping and material growth can create thermoelectric devices with high figure of merit [109].

## 6. Conclusions

Transition metal dichalcogenides have made incredible progress recently in the development of materials and heterostructures for the capabilities of both contact and non-contact thermal sensing in a wide range of temperatures. Several devices have shown broadband photoresponse in visible and infrared ranges by incorporating defects, heterostructures, and photogating in their devices, as well as shown enhanced responsivity, detectivity, and response time with the ability to tune the devices for a wide range of applications. Also, highly accurate and ultrathin thermal sensing is possible using 2D TMD-based thermoelectric devices reported. The tailoring of heterostructures, inorganic/polymer composites, and nanoribbon armchair devices have shown to drastically increase thermoelectric performance, which can lead to industrial or consumer applications.

Though TMDs have shown a wide range of properties which can be useful to temperature sensing, many challenges are ahead in the further development of such devices. For one, the growth of high-quality, tailored materials poses a strong challenge that many groups are working toward resolving. Also, some of the limiting band gap qualities of the material inhibits it to niche applications in high-temperature thermal sensing. Although some groups described in this review show broadband-sensitive TMD devices, there is much more work to do in developing heterostructure devices and sensors. Regardless, mono and few-layer TMDs have shown very promising capabilities in this field for a wide range of applications.

## Figures and Tables

**Figure 1 micromachines-11-00693-f001:**
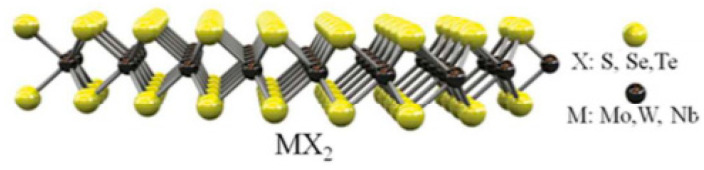
Image of a monolayer transition metal dichalcogenides (TMDs) atomic structure. Reprinted with permission from Reference [22], copyright 2015 Royal Chemical Society.

**Figure 2 micromachines-11-00693-f002:**
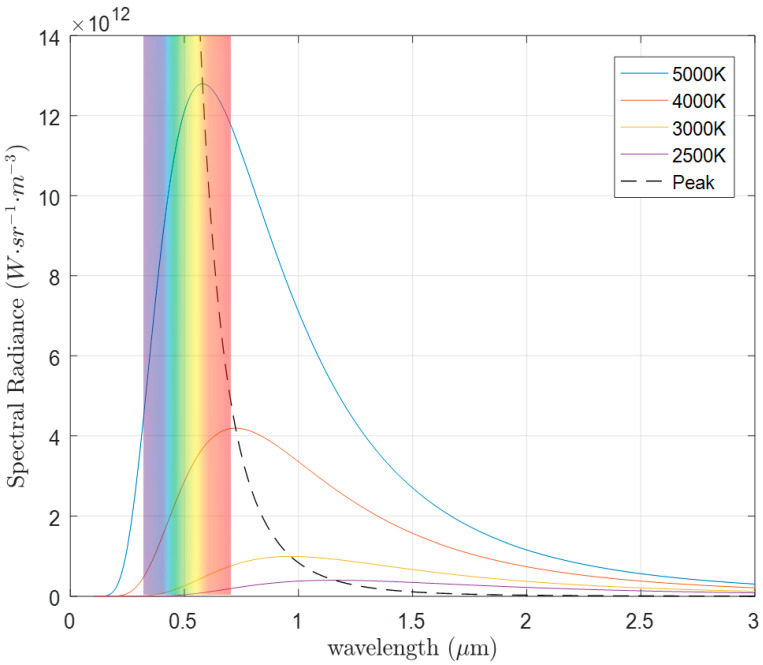
Spectral radiance of a blackbody at various temperatures with a dotted line of peak intensity trend.

**Figure 3 micromachines-11-00693-f003:**
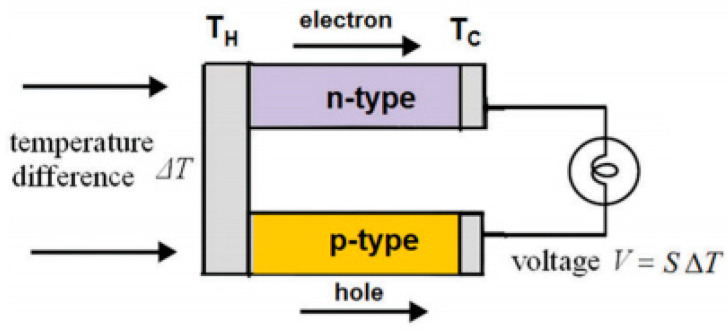
Thermoelectric generator using the Seebeck effect. Republished with the permission of the Royal Society of Chemistry, from Reference [21]. Permission conveyed through Copyright Clearance Center, Inc.

**Figure 4 micromachines-11-00693-f004:**
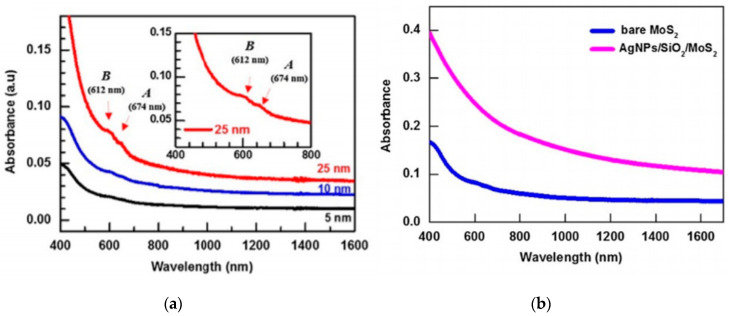
(**a**) Absorbance of 5 nm, 10 nm, and 25 nm thick MoS_2_ from wavelength ranges between 400–1600 nm. Inset image shows a zoomed in view of A and B absorbance peaks for 25 nm thick film. (**b**) Absorbance of 25 nm thick MoS_2_ and AgNP/SiO_2_/MoS_2_ between 400–1600 nm. Reprinted from Reference [48], copyright 2018, with permission from Elsevier.

**Figure 5 micromachines-11-00693-f005:**
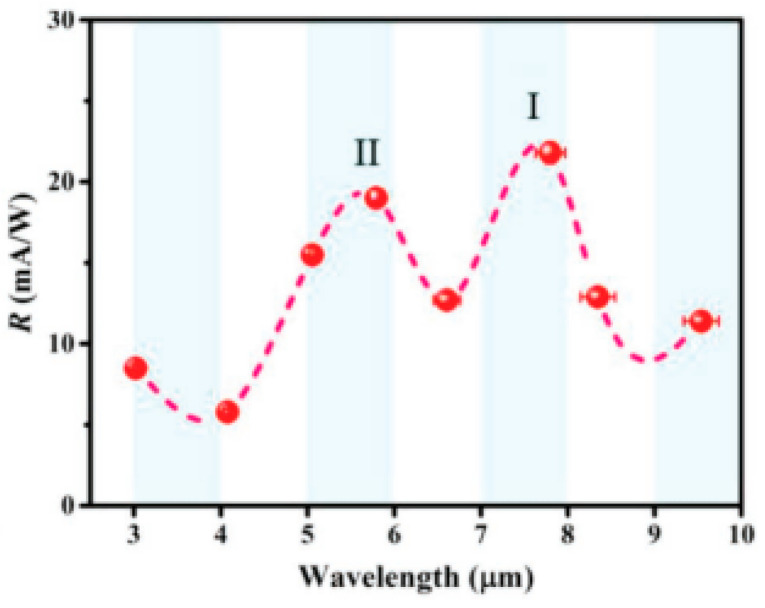
Responsivity with respect to wavelength of defect engineered MoS_2_. Reprinted with permission from Reference [53].

**Figure 6 micromachines-11-00693-f006:**
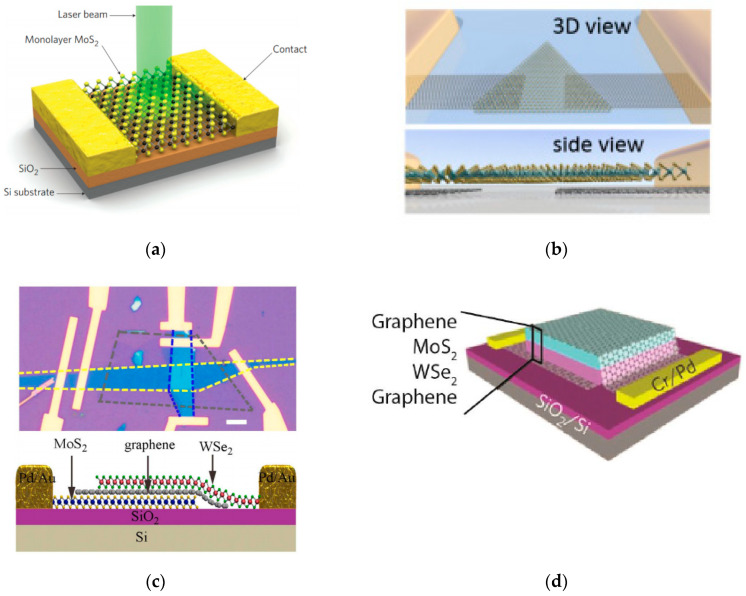
(**a**) Basic 3D monolayer MoS_2_ photodetector schematic with incident laser light source. Reprinted with permission from Springer Nature [37], Copyright 2013. (**b**) 3D and sideview of WS_2_ photodetector with graphene electrodes. Reprinted (adapted) with permission from Reference [54], copyright 2016 American Chemical Society. (**c**) Image of (top) top view of WSe_2_/graphene/MoS_2_ with electrodes and (bottom) cross-sectional view. Reprinted (adapted) with permission from Reference [57], copyright 2016 American Chemical Society. (**d**) MoS_2_/WSe_2_ heterostructure sandwiched between two graphene layers. Reprinted by permission from Reference [64], copyright 2014 Springer Nature Publishing group.

**Figure 7 micromachines-11-00693-f007:**
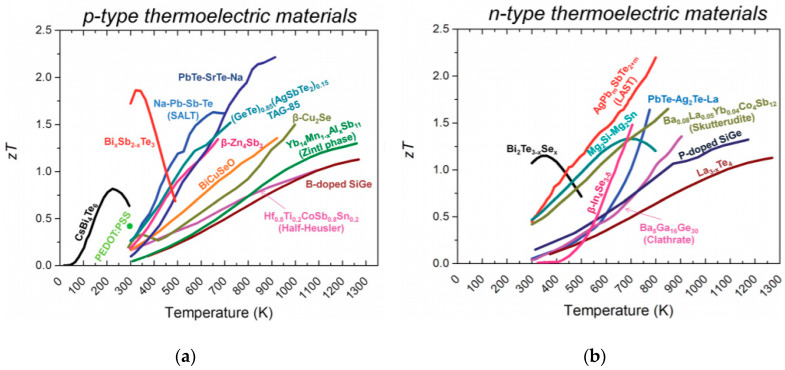
Current bulk figures of merit for (**a**) p-type and (**b**) n-type thermoelectric materials over temperatures. Reproduced with permission from reference [80]. Copyright 2011 Royal Chemical Society.

**Figure 8 micromachines-11-00693-f008:**
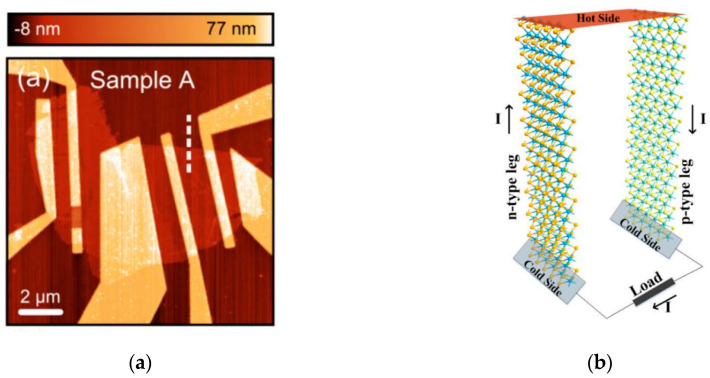
(**a**) Image of fabricated monolayer MoS_2_ photo-thermoelectric generator. Reproduced from [77]. Copyright 2013 American Chemical Society (**b**) Design of an armchair thermoelectric generator using n and p type MoS_2_ nanoribbons. Reproduced from [98] Copyright 2015 Springer Nature.

**Table 1 micromachines-11-00693-t001:** Common blackbody temperatures with their corresponding photon energy, wavelength, and application or phenomenon.

Temperature	Photon Energy Peak (eV)	Peak Wavelength (nm)	Application/Reference
290 K	0.124	9992	Common outside temperature 70 deg F
310 K	0.132	9348	Human body temperature (98 deg F)
373 K	0.16	7769	Water boiling temperature
1300 K	0.56	2229	Flowing lava, open flame
3500 K	1.49	828	Rocket combustion temperature
5800 K	2.47	500 nm	Surface temperature of the Sun

**Table 2 micromachines-11-00693-t002:** Various TMD-based photodetectors and figures of merit.

Material	Layers	Spectral Range (nm)	Responsivity	Detectivity	Response Time (Rise/Fall)	Reference
MoS_2_	1	400–680	880	-	-	[37]
MoS_2_	3	380–800	0.57	10^10^	70/110 µs	[56]
MoS_2_	>40	445–9500	21.8 (At 7790 nm)	-	-	[53]
AgNPs-MoS_2_	~8–40	400–1600	0.881 mA/W	1.28 × 10^9^ J		[48]
MoS_2_2/1L-Gr/WSe_2_	3	400–2400	Visible–10^4^ A/W 2400 nm–~1A/W	Visible- 10^15^ Jones 2400 nm- 10^9^ Jones	30 ms (rise)	[57]
MoTe_2_ (mechanically exfoliated)	2H?	600–1750	24 mA/W	1.3 × 10^9^	-	[50]
1T’ MoTe_2_	4	500–1100	62–109 mA/W	-	0.82 µs (rise) 7.29 (fall)	[58]
Si/MoS_2_ heterojunction	6–10	450–1000	8.75 A/W	1.28 × 10^9^ J	10/19 s	[59]
WS_2_/Si (Type II)	~5	200–3043	224 mA/W	1.5 × 10^12^ J	16/29 µs	[60]
WS_2_/graphite on paper	Nanosheets	390–1080	6.66 mA/W	1.94 × 10^8^ J	800/1400 ms	[61]
p-CuO/n-MoS_2_	1	-	11.4 mA/W	3.27 × 10^8^ J	-	[62]

**Table 3 micromachines-11-00693-t003:** Various TMD-based thermoelectric devices and thermoelectric figures of merit.

Material/Type	Layers	$S^{2}\sigma \;(µ{\mathrm{Wm}}^{-1}K^{-2})$	$S\;(\mu VK^{-1})$	σ (S/cm)	κ (Wm^−1^K^−1^)	ZT	Temperature	Reference
p-MoS_2_ (theoretical)	1	-	-		-	0.55	300 K	[81]
n-WSe_2_ (theoretical)	1	-	-		-	1.1	500 K	[81]
MoS_2_	1	-	−4 × 10^2^ −1 × 10^5^		-	-	-	[85]
n-WSe_2_ and p-MoS_2_	1	>200	>200		1.5 (from [101])	0.1	300 K	[87]
WSe_2_ (gate optimized)	3	3200 (n-type) 3700 (p-type)	~100	~5 × 10^3^	-	-	300 K	[86]
rGO-MoS_2_	-	15.1	~80	130.8	0.206	0.022	300 K	[92]
rGO-WS_2_	-	17.4	~80	136.4	0.208	0.025	300 K	[92]
WS_2_ PEDOT:PSS (50%)	Nano-sheets	45.2	~83	1333	0.36 (cross-plane) 1.2 (in-plane)	0.01	300 K	[95]
MoS_2_/MoSe_2_ (armchair)	Nano-ribbon	-	~600	-	-	7.4	800 K	[100]

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
