# Peer review of "Thermal and Photo Sensing Capabilities of Mono- and Few-Layer Thick Transition Metal Dichalcogenides"

_micromachines, 2020, doi:10.3390/mi11070693_

Round 1

Reviewer 1 Report

This review surveyed the temperature sensing capability of transition metal dichalcogenides (TMDs) and their heterostructures in both contact-based and non-contact schemes. The photoresponse of TMDs in visible and infrared ranges was discussed and the methods to improve responsivity, detectivity, and response time were presented. Also, the performance of TMD-based thermoelectric devices in terms of their figure of merit (ZT), Seebeck coefficient, and power factor were reviewed and several strategies to boost their performance were mentioned. Given 2D materials being one of the hottest research directions and temperature sensing is critical in a broad range of applications, this work serves as a timely contribution and is certainly of interest to the material readership. In order to be worthy published on the Micromachines, some improvements can be made as below,

  1. I believe that the manuscript is not well polished in terms of language. There are long and repeated sentences/words throughout the manuscript that need to be revised. For example on page 10, Part 4. TMD-based thermocouple/ thermoelectric devices, paragraph two and three have similar sentences on properties of graphene.

  1. On page 5, the highest reported ZT is mentioned to be 2.5 in ref. 30. There is a recent paper “Nature 2019, 576, 85–90” that reports a ZT ~ 6 for a thin film Heusler alloy based on Fe2V0.8W0.2Al. I suggest revising this part accordingly.

  1. On page 10, paragraph 3, it is mentioned “ TMDs on the other hand, have shown good thermal conductivity in few layer materials of about 52 Wm-1K-1 for monolayer MoS2 [71].”, and this is followed by several theoretical works on thermoelectric properties of TMDCs. Since experimentally the properties of TMDs depend on many factors such as defect density, material processing, measurement methodology etc It would be informative for the reader to cite more papers including review articles in this section covering this point. For instance : Phys. Rev. B 95, 115407 (2017); Adv. Funct. Mater.27, 1704357 (2017); ACS Appl. Mater. Interfaces2018, 10, 5, 4921–4928; DOI: 10.1073/pnas.2007495117; etc.

There are a few typos as below:

Page 2, last paragraph, “ Figure 1 shows the peak spectra radiance changes with the temperature” should be “ Figure 2 …. “

Page 3, 2nd paragraph, “ as equation 1 and figure 2 show, the peak spectra radiance blue shift to higher wavelengths as temperature increases” should be “ as temperature decreases”.

Page 4, last paragraph, “ Figure 7” should be “ Figure 3"

Author Response

  1. I believe that the manuscript is not well polished in terms of language. There are long and repeated sentences/words throughout the manuscript that need to be revised. For example on page 10, Part 4. TMD-based thermocouple/ thermoelectric devices, paragraph two and three have similar sentences on properties of graphene.
    • Revised paragraph in question and others
  2. On page 5, the highest reported ZT is mentioned to be 2.5 in ref. 30. There is a recent paper “Nature 2019, 576, 85–90” that reports a ZT ~ 6 for a thin film Heusler alloy based on Fe2V0.8W0.2Al. I suggest revising this part accordingly.
    • Revised and included reference
  3. On page 10, paragraph 3, it is mentioned “ TMDs on the other hand, have shown good thermal conductivity in few layer materials of about 52 Wm-1K-1 for monolayer MoS2 [71].”, and this is followed by several theoretical works on thermoelectric properties of TMDCs. Since experimentally the properties of TMDs depend on many factors such as defect density, material processing, measurement methodology etc It would be informative for the reader to cite more papers including review articles in this section covering this point. For instance : Phys. Rev. B 95, 115407 (2017); Adv. Funct. Mater.27, 1704357 (2017); ACS Appl. Mater. Interfaces2018, 10, 5, 4921–4928; DOI: 10.1073/pnas.2007495117; etc.
    • Included several works mentioned and clarified some areas in section, as well as included some review articles.
  4. Fixed typos mentioned and some others throughout manuscript

Reviewer 2 Report

The author reports the recent progress of transition metal dichalcogenides (TMD) for their photodetection, thermoelectric properties and potential application in contact-based (thermocouple), and non-contact (photodetector) thermal sensing. This review focus on TMD’s optical, electrical, and thermoelectric capabilities to be used in thermal sensing. Based on the theory of non-contact and contact temperature sensing, the author introduces them to photoelectric and thermoelectric applications. And put forward key challenges and future efforts. The manuscript is a little disordered and the writing is a bit casual. The author does not show a good combination of photoelectric, thermoelectric and temperature sensing capabilities. These parts are independent relatively. Overall, the manuscript is meaningful for TMD materials in photoelectric and thermoelectric applications.

  1. The author writes ‘Broadband photodetection is a method of temperature sensing with TMDs’ but the photoelectric detection mentioned later has little connection with temperature sensing.
  2. The author writes ‘namely its low thermal conductivity….52 W/mK’, this value seems not a low thermal conductivity.
  3. What is high bandgap semicondutors
  4. Line 36-37 is same as line 30-31
  5. Line 167: 400-680nm seems not a broad spectral range.
  6. Line 338: What means good thermal conductivity.

Author Response

  1. The author writes ‘Broadband photodetection is a method of temperature sensing with TMDs’ but the photoelectric detection mentioned later has little connection with temperature sensing.
    • Attemped to clarify TMD-based photodetectors and their use in temperature sensing for section
  2. The author writes ‘namely its low thermal conductivity….52 W/mK’, this value seems not a low thermal conductivity.
    • Clarified that this thermal conductivity was relative to graphene in the context of the paragraph
  3. What is high bandgap semicondutors
    • clarified band gap of TMDC's (not high, but wide)
  4. Altered paragraphs for clarity and removed repetition
  5. Changed wording of section
  6. Changed wording for context of situation